

# Insight into the Composition of Organic Compounds

# (≥ C₆) in PM₂.₅ in Wintertime in Beijing, China

**Ruihe Lyu[1,2], Zongbo Shi[2], Mohammed Salim Alam[2],**
**Xuefang Wu[2, 4], Di Liu[2], Tuan V. Vu[2], Christopher Stark[2],**
**Pingqing Fu[3], Yinchang Feng[1] and Roy M. Harrison[2]†***

**[1] State Environmental Protection Key Laboratory of Urban Ambient Air
Particulate Matter Pollution Prevention and Control, College of
Environmental Science and Engineering
Nankai University, Tianjin 300350, China**

**[2] Division of Environmental Health and Risk Management
School of Geography, Earth and Environmental Sciences, University of
Birmingham Edgbaston, Birmingham B15 2TT, UK**

**[3] Institute of Surface Earth System Science, Tianjin University
Tianjin, 300350, China**

**[4] Regional Department of Geology and Mineral Resources, China.
University of Geosciences, Xueyuan Road 29, 100083 Beijing China**

**† Also at: Department of Environmental Sciences / Centre of Excellence in
Environmental Studies, King Abdulaziz University, PO Box 80203, Jeddah, 21589,
Saudi Arabia**

**Corresponding author:** E-mail: r.m.harrison@bham.ac.uk (Roy M. Harrison)



## ABSTRACT

Organic matter is a major component of $PM_{2.5}$ in megacities. In order to understand the detailed characteristics of organic compounds ($>C_6$) at a molecular level on non-haze and haze days, we determined more than 300 organic compounds in the $PM_{2.5}$ from an urban area of Beijing in November-December 2016 using two-dimensional gas chromatography coupled to time-of-flight mass spectrometry (GC × GC-TOFMS). The identified organic compounds have been classified into groups, and quantitative methods were used to calculate their concentrations. Primary emission sources make significant contributions to the atmospheric organic compounds and six groups (including n-alkanes, PAHs, levoglucosan, branched-alkanes, n-alkenes and alkyl-benzenes) account for 66% of total identified organic compound mass. In addition, polycyclic aromatic hydrocarbons (PAHs), and oxygenated PAHs (OPAHs) were abundant amongst the atmospheric organic compounds on both haze and non-haze days. A near-unimodal molecular distribution, peaking approximately within the range of $C_{19}$-$C_{28}$, was observed in most hydrocarbon groups. In addition, the concentrations of unidentified compounds were also estimated in the present study. The total identified compounds account for approximately 47% of total organic compounds ($> C_6$) in the chromatogram on both the non-haze and haze days. The total mass concentrations of organic compounds ($> C_6$) in the chromatogram were 4.0 μg m$^{-3}$ and 7.4 μg m$^{-3}$ on the non-haze and haze days respectively, accounting for 26.5% and 18.5% of OM respectively on those days. There is strong evidence that the organic aerosol is more highly oxidised, and hence less GC-volatile on haze days.

**Keywords:** Organic aerosol; GC × GC-TOFMS; PAHs; Haze; $PM_{2.5}$ Beijing, China



## 1.    INTRODUCTION

Organic matter is a large and important fraction of atmospheric fine particles and is composed of

hundreds of organic compounds (Wu et al., 2018). It can influence visibility degradation (Facchini et al.,

1999), affect atmospheric chemical processes, and have a great impact on human health (Bi et al., 2005).

A substantial number of organic compounds can be found in the atmospheric particulate phase and may

originate as either primary emissions or from secondary sources. Due to its huge complexity, particulate

organic matter is still poorly characterized up to the present. In order to establish relationships between

organic compounds in fine particles and their characteristics on non-haze and haze days, as well as to

identify the relative importance of their emission sources, further investigation of particulate organic

matter composition has been conducted.

With growing urbanization, increasing vehicle numbers, rapid economic development, and large energy

consumption, China is suffering from severe $PM_{2.5}$ pollution, especially in its capital, Beijing which has

been experiencing serious air pollution over the past two decades.   The annual average concentration of

$PM_{2.5}$ in Beijing was in the range 69.7–122 μg m$^{-3}$ during the past decade (Lang et al., 2017), 2.0–3.5

times the national standard (35 μg m$^{-3}$). The average $PM_{2.5}$ concentration during the winter period from

November 2014 to March 2015 was 96 μg m$^{-3}$ in Beijing, which was 9.6 times the World Health

Organization (WHO) guideline of 10 μg m$^{-3}$. As a result, $PM_{2.5}$ has received a great deal of attention in

recent studies (Feng et al., 2006; Li et al., 2013; Ren et al., 2016; Yao et al., 2016), and specific tracers

and precursor compounds, including n-alkanals, PAHs, hopanes and alcohol and acid, have been

extensively studied. However, the speciated chemical composition of the urban organic aerosol is far

from complete, with many studies concentrating on structurally specific identifications rather than "group

type".



Two-dimensional gas chromatography (GC×GC) coupled with TOF-MS offers much enhanced
resolution of complex mixtures, and the technique has been extended in the last 10 years to encompass
atmospheric analysis. The three independent analytical dimensions in GC×GC-TOF/MS make this
technique potentially ideal for measuring the organic components within a complex matrix such as
ambient particulate matter (Hamilton et al., 2004; Welthagen et al., 2003), and its ability to separate
complex mixtures of organics at low concentrations makes it an ideal technique to measure partially
oxidised, isomeric and homologous series compounds and even groups of compounds (Alam et al.,
2016a; Alam and Harrison, 2016; Hamilton et al., 2004).  In the earlier study of organic compounds in
the Beijing atmosphere, Zhou et al. (2009) reported that 68.4% of particulate organic matter was in the
previously "unresolved complex mixture" found in conventional GC separations.  The GC × GC
technique is able to resolve and identify the components contributing to the unresolved mixture.

The objective of this study was to investigate the organic compounds with carbon number higher than
$C_6$ in $PM_{2.5}$ samples collected in central Beijing during wintertime, 2016. In this paper, particle samples
were analysed by the GC×GC-TOFMS technique after solvent extraction and the detailed organic
composition was observed for polar and non-polar organic compound groups. Here, we report a large
number of organic compounds, and their concentrations and molecular distributions sampled on non-
haze and haze days. In addition, we report their possible sources, formation processes, and reveal and
assess their pollution characteristics during non-haze and haze periods. Finally, the mass of unidentified
organic compounds ($\geq C_6$) was estimated and compared between non-haze and haze days.








## 2.    MATERIALS AND METHODS

### 2.1    Sampling Method and Site Characteristics

$PM_{2.5}$ samples were collected at the Institute of Atmospheric Physics (IAP), Chinese Academy of Sciences in Beijing, China. The sampling site (89°58′28″ N, 11°62′16″ E) was located between the North 3rd Ring Road and North 4th Ring Road (Figure 1). The site is approximately 1 km from the 3rd Ring Road, 200 m west of the G6 Highway (which runs north-south) and 50 m south of Beitucheng West Road (which runs east-west). The annual average vehicular speeds in the morning and evening traffic peak were 27.4 and 24.3 km $h^{-1}$, respectively. No industrial sources were located in the vicinity of the sampling site. The experimental campaign took place from Nov 9 to Dec 11, 2016. The samples were collected onto pre-baked quartz fibre filters (Pallflex) by a gravimetric high volume sampler (Tisch, USA) with a $PM_{2.5}$ inlet at a flow rate of 1.0 $m^3$ $min^{-1}$ during the sampling period. The collecting time was 24 h per sample and 3 blank samples were collected during this period. The filters were previously enveloped with aluminium foils and then baked at 450 °C for 6 hours before sampling. After sampling, each filter was packed separately and stored in a refrigerator below -20°C until the analysis.

### 2.2    Analytical Instrumentation

The sample extracts were analyzed using a 2D gas chromatograph (GC, 7890A, Agilent Technologies, Wilmington, DE, USA) equipped with a Zoex ZX2 cryogenic modulator (Houston, TX, USA).  Full details of the method appear in Lyu et al. (2018a).  Further details of the instrumentation and data processing methods are given by Alam and Harrison (2016) and Alam et al. (2016a).

### 2.3    Qualitative and Quantitative Analysis

Standards used in these experiments included 26 n-alkanes ($C_{11}$ to $C_{36}$), EPA's 16 priority pollutant polycyclic aromatic hydrocarbons (PAHs), 4 hopanes (17α(H),21β(H)-22R-homohopane,


17α(H),21β(H)-hopane, 17b(H),21a(H)-30-norhopane and 17α(H)-22,29,30-trisnorhopane, 7 decalins
and tetralines (cis/trans-decalin, tetralin, 5-methyltetraline, 2,2,5,7-tetramethyltetraline, 2,5,8-
trimethyltetraline and 1,4-dimethyltetraline), 4 alkyl-naphthalenes (1-methyl-naphthalene, 1-ethyl-
naphthalene, 1-n-propyl-naphthalene and 1-n-hexyl-naphthalene), 15 alkyl-cyclohexanes (n-heptyl-
cyclohexane to n-nonadecyl-cyclohexane), 6 alkyl-benzenes (n-butyl-benzene, n-hexyl-benzene, n-
octyl-benzene, n-decyl-benzene and n-dodecyl-benzene) (Sigma-Aldrich, UK, purity >99.2%), 12 n-
aldehydes ($C_8$ to $C_{13}$) (Sigma-Aldrich, UK, purity ≥95.0%), $C_{14}$ to $C_{18}$ (Tokyo Chemical Industry UK
Ltd, purity >95.0%); and 10 2-ketones, $C_8$ to $C_{13}$ and $C_{15}$ to $C_{18}$ (Sigma-Aldrich, UK, purity ≥98.0%)
and $C_{14}$ (Tokyo Chemical Industry UK Ltd, purity 97.0%), 4 n-alcohols ( 2-decanol, 2-dodecanol, 2-
hexadecanol and 2-nonadecanol) (Sigma-Aldrich, UK, purity 99.0%) and 1-pentadecanol (Sigma-
Aldrich, UK, purity 99.0%).

Compound identification was based on the GC×GC-TOFMS spectral library, NIST mass spectral
library and on co-injection with authentic standards, as described by Lyu et al., (2018a). The calibration
curves for all target compounds were highly linear ($r^2$>0.98, from 0.978 to 0.998), demonstrating the
consistency and reproducibility of this method. Limits of detection for individual compounds were
typically in the range 0.001–0.08 ng m$^{-3}$. The identified compounds which have no commercial
authentic standards were quantified using the calibration curves for similar structure compounds or
isomeric compounds. This applicability of quantification of individual compounds using isomers of the
same compound functionality (which have authentic standards) has been discussed elsewhere and has
a reported uncertainty of 24% (Alam et al., 2018).

The branched alkanes, alkyl-benzenes, alkyl-decalins, alkyl-phenanthrene and anthracene (alkyl-Phe and
Ant), alkyl-naphthalene (alkyl-Nap) and alkyl-benzaldehyde were identified in the samples with the



graphics method of the GC Image v2.5 (Zoex Corporation, Houston, US), and the detailed descriptions
are given elsewhere (Alam et al., 2018). Briefly, the structurally similar compounds (similar physico-
chemical properties) were identified as a group via drawing a polygon around a section of the
chromatogram with the polygon selection tool. All compounds included in the polygon belong to a
special compound class and the total concentrations were calculated via a calibration curve of the
adjacent compounds and IS.

Field and laboratory blanks were routinely analysed to evaluate analytical bias and precision. Blank
levels of individual analytes were normally very low and, in most cases, not detectable. The major
contaminants observed were very minor amounts of n-alkanes ranging from $C_{11}$ to $C_{21}$, with no carbon
number predominance and maximum at $C_{18}$; PAH were not detectable. The major proportion of the
contaminants could be distinguished by their low concentrations and distribution fingerprints (especially
the n-alkanes). These contaminants did not interfere with the recognition or quantification of the
compounds of interest. Recovery efficiencies were determined by analysing the blank samples spiked
with standard compounds. Mean recoveries ranged between 82 and 98%. All quantities reported here
have been corrected according to their recovery efficiencies.  Analytical data from the GC×GC analysis
were compared with a conventional GC-MS analysis for levoglucosan and 13 PAH.  In all cases the
methods correlated moderately to well ($r^2$ = 0.5 to 0.8) with 4 mean concentrations within 18%, 6 within
10-20%, 2 within 20-30% and the remainder (2) within 30-40% of one another. The largest outlier was
levoglucosan, which was underestimated, probably since it decomposed due to a lack of the usual
derivatisation.





## 3.    RESULTS AND DISCUSSION

### 3.1    General Aerosol Characteristics

33 samples were separated as non-haze (13) and haze (20) days (with $PM_{2.5}$ exceeding 75 µg m$^{-3}$ for 24 h average) according to the National Ambient Air Quality Standards of China (NAAQS) released in 2012 by the Ministry of Environmental Protection (MEP) of the People's Republic of China. The concentrations of $PM_{2.5}$, black carbon (BC), organic carbon (OC), element carbon (EC), gaseous pollutants ($SO_2$, NO, $NO_2$, $NO_x$, and CO) and meteorological parameters (wind speed (WS), wind direction (WD) and relative humidity (RH)) were simultaneously determined during the field campaigns and appear in Table S1. Detailed descriptions are given in Shi et al (2018). The average concentration of organic matter (OM) was estimated as 30.2 µg m$^{-3}$ using the OC concentration (18.9 µg m$^{-3}$) and a multiplying factor of 1.6 for aged aerosols (Turpin and Lim, 2001). The OM concentration was 40.0 µg m$^{-3}$ and 15.0 µg m$^{-3}$ on haze and non-haze days respectively.

### 3.2    The Major Classes of Organic Compound in $PM_{2.5}$

More than 6000 peaks were found in the 2D chromatogram image of each sample by the data processing software (GC Image v2.5). Over 300 polar and non-polar organic compounds (POCs and N-POCs) were identified and quantified in the $PM_{2.5}$ samples, and these compounds are grouped into more than twenty classes, including normal and branched alkanes, n-alkenes, aliphatic carbonyl compounds (1-alkanals, n-alkan-2-ones and n-alkan-3-ones), n-alkanoic acids, n-alkanols, polycyclic aromatic hydrocarbons (PAHs), oxygenated PAHs (OPAHs), alkylated-PAHs, hopanes, alkyls-benzenes, alkyl-cyclohexanes, pyridines, quinolines, furanones, and biomarkers (levoglucosan, cedrol, phytane, pristane, supraene and phytone). The details of aliphatic hydrocarbon measurements (including n-alkanes, n-alkenes) and carbonyl compounds (including n-alkanals, n-alkan-2-ones, n-alkan-3-ones, furanones and phytone) have been reported in a previous article (Lyu et al. 2018a,b). The total concentrations of identified organic compounds ranged from 0.94 to 5.14 µg m$^{-3}$ with the average of 2.84 ± 1.19 µg m$^{-3}$, accounting for



9.40 % of OM. The concentrations of identified individual organic compounds are summarized in Table
S2, and the percentage of each group in the total identified organic compounds is in Figure 1. The n-
alkanes (16%) make the greatest contribution to the total mass of identified organic compounds, followed
by levoglucosan (13%), branched-alkanes (13%), PAHs (10%), n-alkenes (7%) and alkyl-benzenes (7%).
These six groups account for 66% of total identified organic compounds by mass.  In a study in Nanjing,
Haque et al. (2018) reported the most abundant classes of organic compounds to be n-alkanes, fatty acids,
PAHs, anydro-sugars, fatty alcohols and phthalate esters.

**3.3      The Characteristics of Organic Compound Groups on Non-haze and Haze Days**
The average total concentration of identified groups was calculated for the non-haze (13 days) and haze
periods (20 days), the latter considered as $PM_{2.5} > 75$ µg m$^{-3}$. The comparisons of two periods (non-haze
and haze days) are shown in Figure 2, and the detailed concentrations of each group are shown in the
Table S3. The concentrations of most organic compound groups on the haze days were higher than non-
haze days, especially for the n-alkanols and n-Cn-cyclohexanes. The alkyl-benzenes, alkyl-
benzaldehydes, monoaromatic compounds and quinoline have approximately similar concentrations on
the non-haze and haze days.

As many compound groups have not been reported in previous studies, and complete data on the relative
abundance of these compounds in various sources are not available at present, it is not yet possible to
calculate source contributions to ambient organic compound concentrations via molecular marker or
mathematical modelling methods. However, several important consistency checks on the potential source
can be performed. In the sections that follow, the literature on the origin of each of these compound
classes is reviewed briefly and the measured compound concentrations are described. Table 1 shows the
comparison of identified organic compounds between the present and previous studies in Beijing.





### 3.3.1 Short chain fatty acids, n-alkanoic acids, fatty alcohols (n-alkanols) and alkanones


The primary anthropogenic sources of saturated n-fatty acids include the combustion of fossil fuels, wood
and organic detritus. The homologues $< C_{20}$ are thought to be derived from meat cooking (Rogge et al.,
1991), fossil fuel combustion (Simoneit, 1985) and microbial sources, while the homologues $> C_{22}$ are
from vascular plant waxes (Simoneit and Mazurek, 1982). Carbon preference index (CPI), defined as the
ratio of total odd carbon number to even carbon number compounds, has been widely used to evaluate
the relative contribution of biogenic organics and anthropogenic emissions (Bray and Evans, 1961).
Simoneit et al. (1991) reported that the n-alkanoic acids ranged from $C_{12}$-$C_{34}$ at a ground-level site in the
suburbs of Beijing, with a total concentration 40-11,000 ng m$^{-3}$ (CPI=7.3), and believed these compounds
were derived mainly from natural sources (Table 1). In addition, these compounds were identified in the
winter PM collected from Peking University (PKU), with $\sum$ n-alkanoic acid ($C_5$-$C_{32}$) of 426 ng m$^{-3}$
(Huang et al., 2006) and $\sum$ n-alkanoic acid ($C_6$-$C_{32}$) of 363 ng m$^{-3}$ (He et al., 2006), respectively. The
studies at the PKU site also found that the n-alkanoic acid homologues showed a similar distribution
pattern in all seasons, suggestive of a stable origin in all seasons, strongly implying a dominant
contribution from fatty acids in cooking emissions as opposed to secondary formation. The study of Sun
et al. (2013) demonstrated that cooking organic aerosol (COA) measured by AMS was an important local
source contributor to OA (16–30 %) at the same IAP site, particularly during non-haze periods. The
average contribution of COA to OA increased to 36% during the non-haze periods, and even went up to
50% at dinner time. The n-alkanoic acids with carbon numbers from $C_6$ to $C_{10}$ were identified in the
PM$_{2.5}$ at lower individual concentrations, and these data have a similar magnitude to a previous study
(Zhou et al., 2009) (Table 1). Consistent results for acids were observed in this study, and the $\sum$ n-
alkanoic acids had a higher concentration on the non-haze days with an average concentration of 36.4 ng
m$^{-3}$, which is higher than 24.6 ng m$^{-3}$ on haze days. Higher molecular weight alkanoic acids are unlikely
to be volatile under the conditions of the chromatography.



N-alkanols have high rate coefficients for reaction with hydroxyl radicals (OH) (MCM,
http://mcm.leeds.ac.uk/MCMv3.2/home.htt), which result in lifetimes ranging from a few hours to 1-2
days and the reaction products include carbonyl compounds (Leif and Simoneit, 1995). Long-chain n-
alkanols are typically found in the waxy portion of leaf surface materials from plants and trees (Rogge
et al., 1998). In aerosols from Malaysia ($C_{12}$-$C_{34}$, CPI=4.9-17.6) (Bin Abas and Simoneit, 1996),
Heraklion (Island of Crete, Greece; $C_{10}$-$C_{32}$, CPI=6.6-11.9) (Stephanou and Stratigakis, 1993), Greece
($C_{10}$-$C_{35}$, CPI=6.8-32.5) (Pio et al., 2001) and CARBOSOL sites ($C_{22}$-$C_{34}$, CPI=3.1-32) (Oliveira et al.,
2007), the typical distributions of n-alkanols with a high CPI reflect a prevailing biogenic origin. In
contrast to the higher molecular weight compounds, the homologues < $C_{20}$ appear to be derived from an
anthropogenic emission source. Biomass burning should be considered as an important source of lower
molecular weight n-alkanols as these compounds have been identified in cereal straw burning emissions
of particulate matter (Zhang et al., 2007), with chain lengths of $C_{14}$-$C_{30}$. Simoneit et al. reported that the
n-alkanols ranged from $C_{14}$-$C_{30}$ from the site in surburban Beijing with the total concentration 1200 ng
$m^{-3}$ (CPI=10.3), and believed these compounds were mainly derived from plant ( > $C_{20}$) wax and
microbial sources ( < $C_{20}$) (Simoneit et al., 1991) (Table 1). In addition, vehicular emissions are
considered as another important source of atmospheric low molecular weight n-alkanols in Beijing.
Reanalysis of samples from our previous diesel engine studies (Alam et al., 2016b) identified 1-
dodecanol, 1-tetradecanol, 1-hexadecanol and 1-octadecanol in the exhaust, at concentrations of 4.03,
5.4, 0.12, 8.20 μg $m^{-3}$, respectively. The engine study set up and exhaust sampling system are given in
detail elsewhere (Alam et al., 2016b). In the present study, 1-alkanols with even-carbon numbers from
$C_{12}$ to $C_{20}$ were identified in the $PM_{2.5}$, which is quite similar to the engine exhaust samples. The average
∑ n-alkanols concentration was 38.5 ng $m^{-3}$, and ∑ n-alkanols had higher concentrations on the haze days
(59.8 ng $m^{-3}$), approximately eight times greater than 8.39 ng $m^{-3}$ on non-haze days. The above results
suggest that n-alkanol formation is more efficient on haze days, even though vehicular emissions appear



to be another important source. In addition, other primary emission sources may make a significant
contribution to these compounds, including from biomass burning.

**3.3.2     Nitrogen-containing organic compounds (N-CC)**
Nitrogen-containing organic compounds have been reported in many previous studies (Rogge et al.,
1994; Rogge et al., 1993b; Schauer et al., 1996; Zhang et al., 2002). Two important sources of N-
containing compounds are biomass burning and atmospheric photochemical reactions. Fan et al. (2018)
found that N-containing compounds were abundant as primary humic-like substances in fine smoke
particles emitted from the combustion of biomass materials (including rice straw, corn straw, and pine
branches) and coal. In the present study, N-containing compounds were identified in the samples,
including heterocyclic compounds (alkyl-pyridines, alkyl-quinolines) and other N-containing
compounds (nitro, amine compounds). The average $\sum$ alkyl-pyridines, $\sum$ alkyl-quinolines and $\sum$ other
N-containing compounds were $17.4 \pm 7.58$, $16.6 \pm 15.0$ and $30.0 \pm 23.1$ ng m$^{-3}$, respectively, and the
average total concentrations of N-containing compounds was 64.0 ng m$^{-3}$, accounting approximately for
0.2% of the OM.

Amino compounds can originate from biomass burning and coal combustion. Zhang et al. (2002)
reported that amino compounds made up a significant portion (23%) of particulate organic nitrogen in
fine particles (PM$_{2.5}$) collected in Davis (California) over a period of one year.  Akyiiz (2008) reported
that amine compounds were abundant in winter fine particulate matter samples compared to the summer
time, and attributed this to the increased emissions from coal-fired domestic and central heating. Our
study found that the average $\sum$ other N-containing compounds was $34.2 \pm 24.6$ ng m$^{-3}$ on the haze days,
somewhat higher than $22.6 \pm 19.4$ ng m$^{-3}$ on non-haze days. The four isomers of dimethyl aniline (2,3-
dimethyl-benzeneamine,  2,4-dimethyl-benzeneamine,  2,5-dimethyl-benzeneamine  and  3,5-dimethyl-



benzeneamine) have similar total concentrations on non-haze (2.09 ng m$^{-3}$) and haze days (3.23 ng m$^{-3}$),
(Table 1). Quinolines are known to occur in crude oils and shale oil (Schmitter et al., 1983; Simoneit et
al., 1971), and were identified in vehicular exhaust (Rogge et al., 1993a). The study found that the non-
haze and haze days also have similar average $\sum$ alkyl-quinolines, with 16.8 ± 16.5 ng m$^{-3}$ and 16.5 ± 14.4
ng m$^{-3}$ respectively. Straight chain alkyl-pyridines (n-Cn-pyridine) were identified in the PM, with
average $\sum$ alkyl-pyridines of 15.3 ± 8.36 ng m$^{-3}$ on the haze days, which is slightly higher than 12.0 ±
6.02 ng m$^{-3}$ on non-haze days. Alkyl-pyridines may be released from proteins and amino acids on
pyrolysis (Chiavari and Galletti, 1992; Hendricker and Voorhees, 1998; Kögel-Knabner, 1997).

Tracers of tobacco smoke, benzoquinoline and isoquinoline have previously been determined in the PM
collected in Beijing, with 3.10 and 0.22 ng m$^{-3}$ respectively (Zhou et al., 2009). These two compounds
were also identified in the present study, with 4.40 and 0.80 ng m$^{-3}$, respectively. Phthalimide was
identified in the PM at 0.91 ng m$^{-3}$, and was previously identified from the PM collected in Guangzhou
and was considered to be derived from cyclization and aromatization reactions of proteins or from
intermediates in the transformation of carboxyl ammonium salts to nitriles (Zhao et al., 2009).

### 3.3.3    Esters
Phthalate esters are organic chemicals that are commonly used in a variety of consumer products and in
various industrial and medical applications, and are predominantly used as plasticizers to improve the
flexibility of polyvinyl chloride (PVC) resins and other polymers. Table 1 shows a comparison of
phthalate esters (DBP, DEP, DEHP) between the present and previous studies in the winter in Beijing; it
seems that the concentrations of phthalate esters have significantly decreased from winter 2006 to 2009
(Wang et al., 2006; Zhou et al., 2009). The present study found that diisodecyl phthalates, DBP and
DEHP were abundant compounds in the ester group with 49.7 ± 43.2, 16.9 ± 15.5 and 16.0 ± 12.6 ng m$^{-}$





$^3$, respectively. The DBP, DEP and DEHP in Beijing were far lower than that in winter in Tianjin (Kong
et al., 2013) and another fifteen cities around China (Li and Wang, 2015; Wang and Kawamura, 2005;
Wang et al., 2006). In addition, the average $\sum$ Ester was $117 \pm 82.1$ ng m$^{-3}$, with $132 \pm 87.1$ and $89.4 \pm$
$70.0$ ng m$^{-3}$ on haze and non-haze days, respectively. Since phthalates are not chemically bound to the
polymeric matrix, they can enter the environment by escaping from manufacturing processes and by
leaching or vaporising from final products (Staples et al., 1997).

**3.3.4    PAH, O-PAH and alkylated-PAHs & OPAHs**
In all, 23 PAHs (2-6 rings), 19 oxygenated PAHs (O-PAHs) and 14 alkylated-PAHs and alkyl-O-PAHs
were determined in the PM$_{2.5}$ samples. The average total polycyclic aromatic compounds (the sum of $\sum$
PAHs, $\sum$ O-PAHs, $\sum$alkylated-PAHs and O-PAHs, alkyl-PHE and ANT and alkyl-NAP) was 569 ng m$^{-}$
$^3$, accounting for 1.88 % of OM.

The distribution of PAHs is shown in Figure 3; the most abundant PAHs were BbF, followed by CHR,
FLT, BaA and PYR. In all samples, the $\sum$ PAHs ranged from 46.7-727 ng m$^{-3}$ with average $281 \pm 176$
ng m$^{-3}$, accounting for 0.93 % of OM. In addition, the average $\sum$ PAHs was 364 ng m$^{-3}$ during haze days,
but only 159 ng m$^{-3}$ on the non-haze days. It should be noted that retene was detected in most samples,
with an average concentration of $14.4 \pm 17.5$ ng m$^{-3}$. It has been suggested that retene predominantly
originates from the combustion of conifer wood (Simoneit et al., 1991).
Nineteen oxygenated PAHs (OPAHs) make up of a class of PAH derivatives that are present in the
atmosphere as a result of direct emission during combustion and secondary formation by homogeneous
and heterogeneous photo-oxidation processes (Keyte et al., 2013; Ringuet et al., 2012). They are also of
scientific interest because they are, typically, found in the secondary organic aerosol (SOA) formed by
photo-oxidation of PAH (Shakya and Griffin, 2010). In urban samples, polycyclic aromatic ketones



(PAK), polycyclic aromatic quinones (PAQ) and polycyclic aromatic furanones (PAF) are typical groups
of compounds (Lin et al., 2015). The average total concentrations of O-PAH measured in this study
(Figure 4) was 67.9 ng m$^{-3}$. The polycyclic aromatic ketones 4,5-pyrenequinone (4.5-PyrQ) (8.75 ng m$^{-3}$)
and 1,6-pyrenequinone (1.6-PyrQ) (7.38 ng m$^{-3}$) were the most abundant compounds during the
sampling campaign. Four O-PAHs have been identified previously at the PKU site in the 2012 heating
season in Beijing (Table 1);  it is notable that the concentration of AQ was up to 108 ng m$^{-3}$,
approximately 20 times that in the present study (5.12 ng m$^{-3}$). As O-PAHs can be formed during
sampling, it is necessary to be very careful in reconciling their presence with specific sources (Pitts et
al., 1980). The average ∑ O-PAHs was 86.5 ng m$^{-3}$ during haze days, but 39.7 ng m$^{-3}$ on the non-haze
days. The ratio of quinone: parent PAH has been used to assess the air mass age (Alam et al., 2014;
Harrison et al., 2016). The average ratios of phenanthraquinone to phenanthrene (PQ:PHE),
anthraquinone to anthracene (AQ:ANT) and benzo(a)anthracene-7,12-quinone to benzo(a)anthracene
(BaAQ:BaA) were 0.37, 1.27, 0.32, respectively. The PQ:PHE, AQ:ANT and BaAQ:BaA ratios were
0.25, 0.88 and 0.26 on the haze days, which were lower than 0.55, 1.92, 0.40 on non-haze days. The
lower ratios on haze days may be explained by further oxidation of the O-PAH.

### 3.3.5    Molecular markers

Hopanes are present in crude oil, rather than being formed in combustion processes (Simoneit, 1985).
Due to their stability, they are valuable tracers of motor vehicle exhaust (Simoneit, 1985; Cass, 1998)
and are also present in emissions from coal combustion (Oros and Simoneit, 2000). The comparison of
hopanes between this study and previous studies in the winter or heating season of Beijing are shown in
Table 1. Hopanes were extensively present in Beijing PM$_{2.5}$ samples, and their carbon numbers ranged
from C$_{27}$ to C$_{32}$, but not C$_{28}$ (Table 2). The average concentration of hopanes in Beijing was 32.7 ± 24.7
ng m$^{-3}$, with 15.2 ± 10.7 ng m$^{-3}$ and 44.6 ± 24.6 ng m$^{-3}$ on non-haze and haze days, respectively.





Previous studies have found that $C_{29}$ (17a(H), 21h(H)-norhopane) was dominant in the hopane series
and consistent with that from coal combustion (He et al., 2006), while $C_{30}$ (17β(H)21α(H)-hopane and
17a(H), 21β(H)-hopane) was similar to $C_{29}$ in the winter time in Beijing and attributed to gasoline and
diesel exhaust (Simoneit, 1985).

Levoglucosan and methoxyphenols from pyrolysis of cellulose and lignin are usually used as unique
tracers for biomass burning in source apportionment models (Schauer and Cass, 2000). Levoglucosan
(1,6-anhydro-$\beta$-D-glucopyranose) has been for a long time employed as the specific molecular marker
for long-range transport of biomass burning aerosol, based on its high emission factors and assumed
chemical stability (Fraser and Lakshmanan, 2000; Simoneit et al., 2000). It is a highly abundant
compound and the concentrations in winter in Beijing have a significant fluctuation (Table 1). The
average $\sum$ levoglucosan was $355 \pm 232$ ng m$^{-3}$ during the entire sampling period, and $417 \pm 223$ ng m$^{-3}$
in haze episodes, approximately twofold that of the non-haze days, $238 \pm 193$ ng m$^{-3}$, indicating a
significant impact of biomass burning upon wintertime aerosols in Beijing.

Methoxyphenols are usually also considered as tracers for wood burning (Simpson et al., 2005; Yee et
al., 2013) with the average $\sum$ Methoxyphenols $7.29 \pm 7.11$ ng m$^{-3}$, and the haze days ($9.03 \pm 7.93$ ng m$^{-3}$)
twofold greater than non-haze days ($4.74 \pm 4.95$ ng m$^{-3}$) during the campaigns. In Beijing and its
surrounding areas, harvest occurs in late September to October for corn, and biomass fuels are used for
cooking and heating purpose in the winter. However, the methoxyphenols are abundant components in
the smoke from broad-leaf tree and shrub burning (Wang et al., 2009), and have been identified in all
coal smoke (Simoneit, 2002a), so cannot be used as source-specific markers for biomass burning.

Phenolic compounds from the thermal degradation of lignin have been proposed as potentially useful



tracers for wood smoke, and many of them are emitted in relatively high quantities and are specific to
wood combustion sources (Simoneit, 2002b; Simoneit et al., 2004). Another important source of phenolic
compounds is oxidation of monoaromatic and PAHs (Pan and Wang, 2014). Phenols and naphthalenol
were identified in the PM$_{2.5}$, with the average $\sum$ phenolic compounds $21.6 \pm 17.0$ ng m$^{-3}$, with $14.0 \pm$
13.2 ng m$^{-3}$ and $25.9 \pm 17.9$ ng m$^{-3}$ on the non-haze and haze days, respectively. However, it is notable
that the concentrations of naphthalenol identified in the present study were far lower than that of previous
studies (Table 1).

Pristane (Pr) and phytane (Ph) are present in the exhaust of petrol and diesel engines and in lubricating
oil, indicating an origin from petroleum (Simoneit, 1984). They have been observed in the atmosphere
(Bi et al., 2002; Andreou and Rapsomanikis, 2009) and since their presence is ubiquitous in vehicle
exhaust and negligible in contemporary biogenic sources in urban environments, they can be used as
petroleum tracers for airborne particulate matter. The mean values of Pr and Ph in our samples are 2.24
and 1.94 ng m$^{-3}$, respectively. Previous studies have used Pr/Ph ratios as an indicator of biogenic material,
which is indicated by a Pr/Ph ratio far higher than 1.0 (Oliveira et al., 2007), while values close to 1
indicate a petrochemical source (Oliveira et al., 2007; Andreou and Rapsomanikis, 2009). The average
Pr/Ph ratios were 1.15 for PM$_{2.5}$ samples, and this finding is quite similar to the results from the southern
Chinese city of Guangzhou, 1.1-1.8 (Bi et al., 2002), but almost four times greater than Beijing summer
samples (0.3) (Simoneit et al., 1991). The high Pr/Ph indicated that the hydrocarbons in urban aerosol
derive mainly from petroleum residues probably deriving from vehicular emissions in Beijing.

**3.4    The Molecular Distributions of Aliphatic Hydrocarbons**
Figure 4 shows the molecular distributions of aliphatic hydrocarbons on non-haze and haze days. The
total concentrations of branched alkanes (C$_{12}$-C$_{36}$) ranged from 125-647 ng m$^{-3}$ with the average $356 \pm$





173 ng m$^{-3}$ during the sampling period. The average branched alkanes concentration was $440 \pm 144$ ng
m$^{-3}$ during all haze episodes, which was higher than $234 \pm 138$ ng m$^{-3}$ on the non-haze days. The most
abundant branched alkanes were observed at $C_{22}$, with the average concentration of 29.2 ng m$^{-3}$. There
is a clear almost unimodal distribution from $C_8$ to $C_{36}$, most clear in the range of $C_{19}$-$C_{28}$. Similar
distributions were observed for branched and straight chain alkanes in the range of $C_{19}$-$C_{28}$ during the
sampling campaigns. In addition, the branched alkanes make a higher contribution to atmospheric
organic compounds in the range of $C_{19}$-$C_{28}$ on the haze days in contrast to the non-haze days. However,
minor differences were observed in two periods for these compounds with lower carbon numbers ($< C_{19}$),
and showing a higher concentration than n-alkanes during the sampling campaigns. In addition, the ratios
of normal/branched alkanes ($C_{12}$-$C_{36}$) was calculated and ranged from 0.04 to 2.15 (average 0.87) and
0.07-1.97 (average 1.05) on the non-haze days and haze days, respectively. It is difficult to identify the
potential sources of branched alkanes from the literature, although Alam et al. (2016b) reported that
branched alkanes ($C_{11}$-$C_{33}$) were an abundant compound group in diesel exhaust. The increase of high
molecular weight branched alkanes ($> C_{19}$) from non-haze days to haze days is consistent with a primary
emission source, probably linked to coal combustion or vehicular emissions.  The fact that both n-alkanes
and branched alkanes increase quite similarly between non-haze and haze conditions is consistent with a
common source.

Other groups of aliphatic and alicyclic compounds identified in the PM$_{2.5}$, include alkyl-decalins, alkyl-
pyridines, alkyl-furanones, alkyl-cyclohexanes and alkyl-benzenes. Figure 5 shows the molecular
distributions of these series of compounds. Engine studies (Alam et al., 2016b) have also found that
compounds observed in vehicle exhaust beside n-alkanes and PAHs, include straight and branched
cyclohexanes ($C_{11}$-$C_{25}$), various cyclic aromatics, alkyl-decalins and alkyl-benzenes. The particle-bound
n-Cn-cyclohexanes with carbon numbers from $C_{12}$ to $C_{26}$ were identified in diesel exhaust (Alam et al.,



2016b) with a dominant range $C_{18}$-$C_{25}$, and the total (particle + gas) concentration of n-$C_n$-cyclohexanes
was 2.05 µg m$^{-3}$. The n-Cn-cyclohexanes ($C_{20}$-$C_{30}$) were identified at the IAP site with average $\sum$ n-$C_n$-
cyclohexane 39.4 ± 37.1 ng m$^{-3}$. The most abundant range was observed at $C_{22}$-$C_{27}$, highly consistent
with the engine study, implying a significant contribution from vehicle emissions. In addition, the
average $\Sigma$n-$C_n$-cyclohexane ($C_{20}$-$C_{30}$) was 53.3 ± 39.3 ng m$^{-3}$ during haze episodes, approximately five
times higher than 10.8 ± 8.22 ng m$^{-3}$ in the non-haze period, a larger ratio than for other primary
emissions. The alkyl-decalins and tetralin are products obtained by hydrogenation of naphthalene and its
derivatives during the refining process and have been identified in vehicle exhaust (Afzal et al., 2008;
Alam et al., 2016b; Ogawa et al., 2007). The average $\sum$ alkyl-decalins was 110 ng m$^{-3}$, with 85.4 ± 65.5
and 126 ± 110 ng m$^{-3}$ on non-haze and haze days respectively. The $\Sigma$ n-$C_n$-benzene ($C_{16}$-$C_{25}$) identified
in the samples ranged from 7.71 to 410 ng m$^{-3}$ with an average of 56.6 ± 73.0 ng m$^{-3}$. The average $\Sigma$ n-
$C_n$-benzene ($C_{16}$-$C_{25}$) was 77.2 ± 88.2 ng m$^{-3}$ during haze episodes, approximately four times the 23.3 ±
15.1 ng m$^{-3}$ of the non-haze period. Other alkyl-benzenes ($C_9$-$C_{25}$) were also identified and have higher
concentrations at $C_{12}$, especially for the non-haze days.

**3.5**        **The Estimation of Unidentified Compounds**
The estimation method for unidentified compounds is detailed in the Supporting Information. Briefly,
the chromatography image was separated into seven parts according to the main chemical and physical
properties of the organic compounds and the distribution of internal standards (IS), and the detailed
protocol is shown in Table S4. The diagram of the separated image with seven parts is shown in Figure
6a, and the concentrations measured in each part are shown in Figure 6b and Table S5. For the non-haze
days, Section 1 has the highest concentration of 546 ± 406 ng m$^{-3}$, followed by Section 7 (440 ± 312 ng
m$^{-3}$), accounting for 25.8 % and 20.8 % of the total unidentified compounds respectively, implying that
both low molecular weight hydrocarbons (Section 1) and PAHs (Section 7) were the main contributor to





the analysed components of atmospheric particulate matter, probably linked to vehicular emissions and
coal combustion. The concentrations in all sections increased from non-haze to haze days, and Section 5
which contained oxidized monoaromatic compounds has the highest concentrations on the haze days
($985 \pm 707$ ng m$^{-3}$), increased more than three times on the haze days in contrast to non-haze days (289
$\pm$ 184 ng m$^{-3}$). In addition, increased quantities were also found for Section 6 (mainly containing
naphthalene derivatives) and Section 7 (PAHs containing more than two benzene rings), increasing 2.9
and 1.8 times on the haze days in contrast to non-haze days, respectively. In the chromatogram (Figure
6a), volatility decreases from left to right and polarity increases from bottom to top. Hence the main
difference between haze and non-haze days attaches to Sections 5, 6 and 7 (Figure 6b) indicating a more
polar aerosol during periods of haze, consistent with the greater elevation in oxidized monoaromatic
compounds.

For the non-haze days, the sum of identified organic compounds (IOC) with carbon numbers higher than
$C_6$ was 1.85 μg m$^{-3}$, accounting for 46.6 % of total organic compounds. The IOC of the haze days was
almost two times that of non-haze periods, with an average of 3.45 μg m$^{-3}$, accounting for 46.5% of total
measured organic matter. In addition, the sum of unidentified compounds increased from 2.12 μg m$^{-3}$ on
non-haze days to 3.96 μg m$^{-3}$ on haze days, accounting for 53.4 % and 53.5% of total measured organic
matter, respectively. Hence there is no marked difference in the proportions of identified and unidentified
compounds between haze and non-haze conditions.

**3.6     Elevation of Primary and Secondary Constituents during Haze Events**
By definition, concentrations of PM$_{2.5}$ are elevated during haze events, but the question arises as to
whether primary or secondary organic compounds make a larger contribution to the rise in
concentrations. Constituents that are expected to be primary are typically elevated in mean concentration




by a factor of around two (Table S3). Examples are n-alkanes (ratio of haze : non-haze of 2.2),
levoglucosan (1.8) and hopanes (2.9). This is consistent with the ratios for primary gaseous emissions,
including $SO_2$ (ratio of 2.6), CO (2.5) and $NO_x$ (2.2) (Table S1). Surprisingly, however, both BC (ratio
of 3.8) and EC (5.1) (Table S1) are primary constituents with a large haze:non-haze ratio, comparable to
that of $PM_{2.5}$ mass (4.0). Consequently the factors leading to an elevation of concentrations during the
haze appear complex and are likely to be resolved fully only by chemistry-transport models. The
aliphatic carbonyls, which have both primary and secondary sources (Lyu et al., 2018a,b) range from
ratios of 1.6 (n-alkanals) to 2.8 (n-alkan-2-ones). These compounds are quite readily oxidised, and a low
ratio may reflect a high degree of processing to form more oxidised species on the haze days. There are
no compounds in Table S3 certain to be exclusively secondary. However, the results in Figure 6 show an
appreciable elevation in more polar compounds (upper part of the chromatogram) on haze days,
suggestive of a greater relative abundance of more oxidised, possibly secondary compounds in the haze.
The ratio of average $PM_{2.5}$ mass between haze and non-haze days was 4.0, and organic carbon, 2.7 (Table
S1). The ratio for organic matter would be greater than 2.7, due to a higher OM/OC ratio in secondary
compounds. This is strongly suggestive of a greater contribution from an elevation in secondary than
primary species concentrations during the haze events, and that much of the mass lies outside of the
chromatogram due to the low volatility of the secondary species.

**4.    CONCLUSIONS**
Over 300 polar and non-polar organic compounds were determined in the fine particle samples from
Beijing, and these compounds have been grouped into more than twenty classes, including normal and
branched alkanes, n-alkenes, aliphatic carbonyl compounds (1-alkanals, n-alkan-2-ones and n-alkan-3-
ones), n-alkanoic acids, n-alkanols, polycyclic aromatic hydrocarbons (PAHs), oxygenated PAHs
(OPAHs), alkylated-PAHs & O-PAHs, hopanes, n-$C_n$-benzene, alkyls-benzenes, n-$C_n$-cyclohexane,



pyridines, quinolines, furanones, and biomarkers (levoglucosan, cedrol, phytane, pristane, supraene and
phytone). The total concentrations of identified organic compounds ranged from 0.94 to 5.14 μg m$^{-3}$ with
an average of 2.84 ± 1.19 μg m$^{-3}$, accounting for 9.40 % of OM mass. The six groups which accounted
for 66% of total identified organic compound mass included n-alkanes, levoglucosan, branched-alkanes,
PAHs, n-alkenes and alkyl-benzenes, and these were significantly impacted by primary emission sources.
In addition, the average total polycyclic aromatic compounds (the sum of $\sum$ PAHs, $\sum$ O-PAHs,
$\sum$alkylated-PAHs and O-PAHs, alkyl-PHE and ANT and alkyl-NAP) was 560 ng m$^{-3}$, accounting for
1.88 % of OM. The comparisons of identified groups between non-haze and haze periods showed that
most organic compound groups have a higher concentration on the haze days relative to the non-haze
days. The sum of the identified compounds increased from 1.85 μg m$^{-3}$ to 3.45 μg m$^{-3}$ from non-haze
days to haze days. A unimodal molecular distribution of alkanes was observed in the range from C$_8$ to
C$_{36}$, and these compounds make significant contributions to atmospheric organic compounds in the range
of C$_{19}$-C$_{28}$, especially on the haze days. The unidentified compounds in the chromatogram were
estimated, and the results show that the average sum of unidentified compounds increased from 2.12 μg
m$^{-3}$ on non-haze days to 3.96 μg m$^{-3}$ on haze days, accounting approximately for 53.4 % and 53.5% of
total organic compounds, respectively. Finally, the total mass concentrations of measured organic
compounds ($\geq$ C6) was 3.97 μg m$^{-3}$ and 7.41 μg m$^{-3}$ on the non-haze and haze days, accounting for 26.5%
and 18.5% of OM mass, respectively on these days. The remaining mass is that which is not volatile
under the conditions of the gas chromatography. The higher percentage of non-GC-volatile organic
matter on haze days is indicative of a greater degree of oxidation of the organic aerosol, consistent with
the difference in the chromatogram between haze and non-haze days. The greater contribution of
secondary constituents during haze events has been reported previously by Huang et al. (2014) and Ma
et al. (2017), but not the greater extent of oxidation of organic matter. In a modelling study, Li et al.



(2017) found that during winter haze conditions in Beijing the majority of secondary PM$_{2.5}$ had formed
one or more days prior to arrival, hence explaining its highly oxidised condition.

**ACKNOWLEDGEMENTS**
Primary collection of samples took place during the APHH project in which our work was funded by the
Natural Environment Research Council (NERC) (NE/N007190/1). The authors would also like to thank
the China Scholarship Council (CSC) for support to R.L.

**AUTHOR CONTRIBUTIONS**
The study was conceived by RMH and ZS and the fieldwork was organised and supervised by ZS and
PF.  TV and DL undertook air sampling work and general data analyses for the campaign while RL
carried analytical work on the Beijing samples under the guidance of MSA and CS.  XW contributed
analyses of data from London.  RL produced the first draft of the manuscript with guidance from YF and
RMH and all authors contributed to the refinement of the submitted manuscript.






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



**TABLE LEGENDS:**

**Table 1:** Comparison of identified organic compounds with earlier studies in Beijing. Data from the present study are mean ± s.d. for n = 33 samples.

**Table 2:** Molecular formula, diagnostic ions and average concentrations of hopanes identified in $PM_{2.5}$.

**FIGURE LEGENDS:**

**Figure 1:** The percentages of the organic compound groups in the total identified organic compounds.

**Figure 2:** A comparison of organic compound groups between non-haze and haze days. The average total concentration of the identified group was calculated in the non-haze (13 days) and haze periods (20 days), respectively.

**Figure 3:** The distribution of concentrations of PAHs.

**Figure 4:** The molecular distributions of aliphatic hydrocarbons and other homologous series, including n-alkanes, branched alkanes, n-alkenes, carbonyl compounds (n-alkanals, n-alkan-2-ones, n-alkan-3-ones), n-alkanoic acid and alkanols on haze and non-haze days.

**Figure 5:** The molecular distributions of n-$C_n$-cyclohexane, alkyl-bicyclic-alkanes, alkyl-benzenes, n-$C_n$-benzenes, alkyl-furanones and alkyl-pyridines on haze and non-haze days.

**Figure 6:** The separated chromatogram from the GC × GC-TOFMS. (a) the diagram of the separated image with seven parts; (b) The concentration results of unknown organic compounds in each chromatography image part during non-haze and haze days.





**Table 1:** Comparison of identified organic compounds with earlier studies in Beijing. Data from the present study are mean ± s.d. for n = 33 samples.

| Compound name | Concentrations, ng m-3 | |
| --- | --- | --- |
| | Present | Previous study |
| **n-alkanols** | | |
| 1-Dodecanol | 2.27±1.49 | 0.90 j; |
| 1-Tetradecanol | 24.2±88.9 | 3.00 j; |
| 1-Hexadecanol | 6.66±20.7 | 1.2 d; 6.30 j; |
| 1-Octadecanol | 1.69±1.65 | 3.1 d; 20.1 j; |
| 1-Eicosanol | 3.71±2.96 | 19.5 j; |
| | | $\sum$ n-alkanols ($C_{14}$-$C_{30}$) = 1200 e; |
| **n-alkanoic acids** | | |
| Hexanoic acid | 1.80±1.54 | 30.4 i; 0.00 j; |
| Heptanoic acid | 0.73±1.05 | 0.62 j; |
| Octanoic acid | 2.97±2.56 | 29.6 i; 0.62 j; |
| Nonanoic acid | 1.23±1.37 | 2.07 j; |
| Decanoic acid | 22.8±25.2 | 6.4 d; 5.8 i; 1.24 j; |
| | | $\sum$ n-alkanoic acid (C12-C34) = 40-11000 e; $\sum$ n-alkanoic acid (C5-C32) = 426 g; $\sum$ n-alkanoic acid (C6-C22) = 363 h; |
| **Hopanes** | | |
| 18α(H)22,29,30-trisnorneohopane | 2.91±3.06 | 0.22 j; |
| 17α(H)-22,29,30-Trisnorhopane | 1.56±2.74 | 2.75 a; 2.3 d; 0.5 i; 0.21 j; |
| 17α(H)21β(H)-30-norhopane | 9.92±7.63 | 7.19 a; 4.1 d; |
| 17β(H)21α(H)-hopane(moretane) | 5.77±6.12 | 1.32 j; 1.9 d; |
| 17α(H)21β(H)-hopane | 3.71±5.49 | 3.51 a; 3.2 d; 0.8 i; 1.54 j; |
| 17α(H)21β(H)-homohopane(22R) | 1.32±1.31 | 0.63 a; 1.2 d; 0.42 j; |
| 17α(H)21β(H)-homohopane(22S) | 0.83±0.93 | 2.94 a; 1.2 d; 0.63 j; |
| 17α(H),21β(H)-bishomohopane(22S) | 5.23±6.51 | 0.7 d; |
| 17α(H)21β(H)-bishomohopane(22R) | 1.41±1.73 | 0.7 d; |
| **Subtotal** | **32.7±24.7** | |
| **PAHs** | | |
| Naphthalene (NAP,2-rings) | 6.03±4.52 | 0.22 b; 2.4 i; |
| Acenaphthylene (ACY, 2-rings) | 12.7±9.93 | 0.065 b; 0.3 i; |
| Acenaphthene (ACE, 2-rings) | 6.04±8.94 | 0.79 b; 0.3 i; |
| Fluorene (FLU, 3-rings) | 16.6±13.0 | 1.18 b; 0.5 i; 15.6 j; |
| Phenanthrene (PHE, 3-rings) | 8.59±8.49 | 14.0 b; 0.9 d; 1.1 e; 21.65 f; 0.9 i; 95.7 j; |
| Anthracene (ANT, 3-rings) | 6.14±6.53 | 1.70 b; 3.3 d; 0.2 i; 52.3 j; |
| Pyrene (PYR, 4-rings) | 18.9±18.2 | 22.3 b; 12 d; 0.58 e; 31.3 f; 1.0 i; 235 j; |
| Fluoranthene (FLT, 4-rings) | 21.0±20.4 | 41.5 b; 11 d; 0.23 e; 31.8 f; 1.1 i; 222 j; |
| Chrysene (CHR, 4-rings) | 25.5±19.3 | 21.8 b; 1.00 d; 1.00 e; 50.6 f; 1.3 i; 140 j; |
| Benz[a]anthracene (BaA, 4-rings) | 17.6±14.6 | 23.5 b; 19 d; 43.4 f; 0.8 i; 62.9 j; |





| Compound name | Concentrations, ng m-3 | |
| --- | --- | --- |
| | **Present** | **Previous study** |
| Benzo[k]fluoranthene (BkF, 4-rings) | 8.81±7.68 | 17.0 b; 8.3 d; 0.7 i; 30.5 j; |
| Cyclopenta[cd]pyrene (CcP, 5-rings) | 8.60±10.2 | 68.0 j; |
| Perylene (PER, 5-rings) | 3.20±2.69 | 2.81 b; 14 d; 0.2 i; |
| Benzo[b]fluoranthene (BbF, 5-rings) | 38.5±31.8 | 34.0 b; 59 d; 33.1 f; 2.3 i; 134 j; |
| Benzo[a]pyrene (BaP, 5-rings) | 13.1±13.8 | 14.6 b; 14 d; 0.08 e; 40.2 f; 1.1 i; 41.3 j; |
| Indeno[1,2,3-cd]pyrene (IcdP, 6-rings) | 12.3±8.82 | 18.1 b; 15.2 d; 0.32 e; 40.9 f; 1.2 i; 18.2 j; |
| Benzo[ghi]perylene (BghiP, 6-rings) | 12.4±11.1 | 12.2 b; 12 d; 0.33 i; 2.6 i; 59.0 j; |
| Benzo[e]pyrene (BeP, 5-rings) | 15.4±10.3 | 12.4 b; 12 d; 0.65 e; 1.3 i; 72.6 j; |
| Dibenzo [a,h]pyrene (DBA, 5-rings) | 5.68±7.35 | 2.01 b; 3.1 d; |
| Benzo[ghi]fluoranthene ( BghiF,5-rings) | 15.1±15.8 | 0.08 e; 15.3 f; |
| **O-PAHs** | | |
| Anthracenedione (AQ) | 5.12±5.97 | 108 b; |
| 7,12-Benz[a]anthracenequinone (BaAQ) | 4.09±3.61 | 2.14 b; |
| Aceanthrenequinone (AceAntQ) | 2.41±2.89 | 0.01b; |
| Phenanthraquinone (PQ) | 1.45±1.08 | 0.13 b; |
| **Alkylated-PAHs and Alkylated-OPAHs** | | |
| Pyrene, 1-methyl- (1-MePYR) | 21.5±21.5 | 3.80 b |
| Phenanthrene, 1-methyl- (1-MePHE) | 5.29±5.38 | 4.29 b |
| Retene | 5.39±9.72 | 0.12 e; 0.5 i; |
| Dibutyl phthalate (DBP) | 16.9±15.5 | 21 d; 3.00 j; |
| Diethyl Phthalate (DEP) | 2.67±2.91 | 3.5 d; 24.0 j; |
| Di(2-ethylhexyl)-phthalate (DEHP) | 16.0±12.6 | 130 d; |
| Diisobutyl phthalate | 49.7±43.2 | 22 d; |
| Dimethyl phthalate | 2.58±2.80 | 1.5 d; |
| **Ester** | | |
| Dibutyl phthalate (DBP) | 16.9±15.5 | 21 d; 3.00 j; |
| Diethyl Phthalate (DEP) | 2.67±2.91 | 3.5 d; 24.0 j; |
| Di(2-ethylhexyl)-phthalate (DEHP) | 16.0±12.6 | 130 d; |
| Diisobutyl phthalate | 49.7±43.2 | 22 d; |
| Dimethyl phthalate | 2.58±2.80 | 1.5 d; |
| **Biomarkers** | | |
| Levoglucosan | 355±232 | 310 a; 790.3 c; 171 d; 78 h; 97.1 i; 830 j; |
| Phytone | 14.7±11.7 | 0.9 j; |
| Phytane | 1.94±1.05 | 2.3 i; 1.30 j; |
| Pristane | 2.24±1.69 | 1.8 i; 0.67 j; |
| **Other nitrogen compounds (Nitro,amine,heterocyclic compounds)** | | |
| Benzo[f]quinoline | 4.40±4.66 | 3.10 j; |
| Isoquinoline | 0.80±0.83 | 0.22 j; |



| Compound name | Concentrations, ng m-3 | |
| --- | --- | --- |
| | **Present** | **Previous study** |
| **Phenolic compounds** | | |
| 1-Naphthalenol (1-OH-NAP) | 1.56±5.61 | 219 b |
| 2-Naphthalenol (2-OH-NAP) | 1.15±1.21 | 2739 b |
| 2-Dibenzofuranol (2-OHDBF) | 1.84±2.09 | 1469 b |

a.   Beijing, PKU, Heating seasons (Ma et al., 2018);
b.   Beijing, PKU, Heating seasons (Lin et al., 2015);
c.   Beijing, China University of Geosciences (Beijing), winter (Shen et al., 2018);
d.   Beijing, winter of 2003 (Wang et al., 2006)
e.   Beijing, urban, June (Simoneit et al., 1991);
f.   Beijing, urban, haze period (Gao et al., 2016);
g.   Beijing, PKU, winter (Huang et al., 2006);
h.   Beijing, PKU, winter (He et al., 2006)
i.   During the 2008 Beijing Olympic Games, PKU sites, (Guo et al., 2013);
j.   Beijing, urban, winter (Zhou et al., 2009);






**Table 2:** Molecular formula, diagnostic ions and average concentrations of hopanes identified in PM$_{2.5}$.

| Compounds | | Molecular formula | Diagnostic ions | IAP, ng m$^{-3}$ |
|---|---|---|---|---|
| 18α(H)22,29,30-trisnorneohopane | Ts | C$_{27}$H$_{46}$ | 191/370 | 2.91 ± 3.06 |
| 17α(H)-22,29,30-Trisnorhopane | Tm | C$_{27}$H$_{46}$ | 191/370 | 1.56 ± 2.74 |
| 17α(H)21β(H)-30-norhopane | 29αβ | C$_{29}$H$_{50}$ | 191/398 | 9.92 ± 7.63 |
| 17β(H)21α(H)-hopane(moretane) | 30βα | C$_{30}$H$_{52}$ | 191/412 | 5.77 ± 6.12 |
| 17α(H)21β(H)-hopane | 30αβ | C$_{30}$H$_{52}$ | 191/412 | 3.71 ± 5.49 |
| 17α(H)21β(H)-homohopane(22R) | 30αβ-22R | C$_{31}$H$_{54}$ | 191/426 | 1.32 ± 1.31 |
| 17α(H)21β(H)-homohopane(22S) | 30αβ-22S | C$_{31}$H$_{54}$ | 191/426 | 0.83 ± 0.93 |
| 17α(H),21β(H)-bishomohopane(22S) | 30αβ-22S | C$_{32}$H$_{56}$ | 191/440 | 5.23 ± 6.51 |
| 17α(H)21β(H)-bishomohopane(22R) | 30αβ-22R | C$_{32}$H$_{56}$ | 191/440 | 1.41 ± 1.73 |





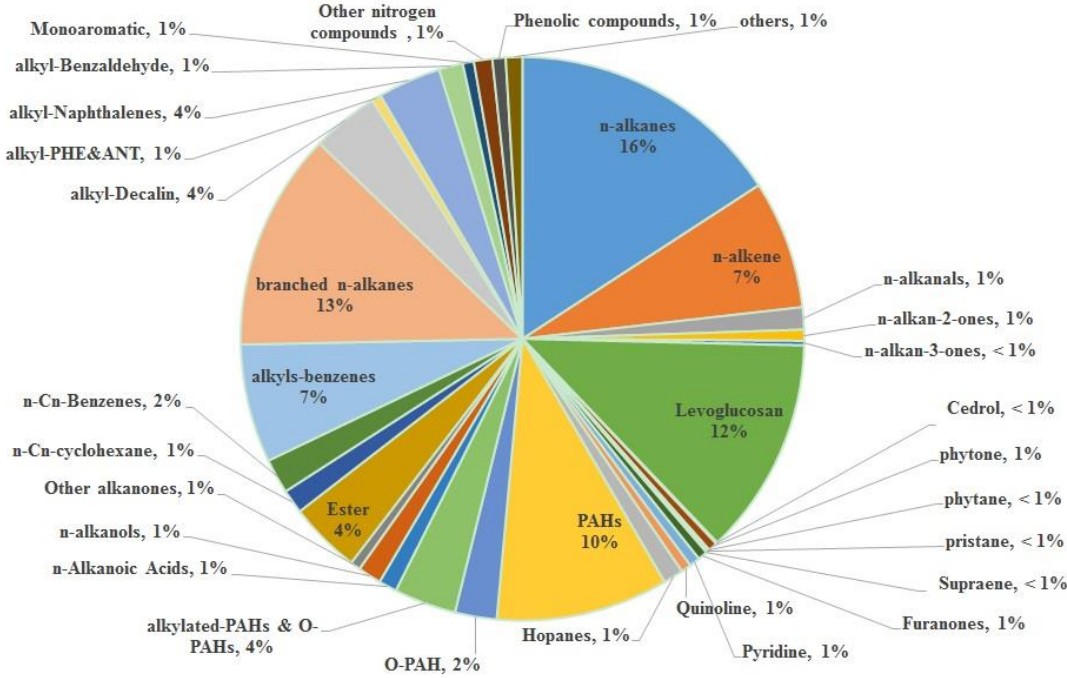


**Figure 1:** The percentages of the organic compound groups in the total identified organic compounds.






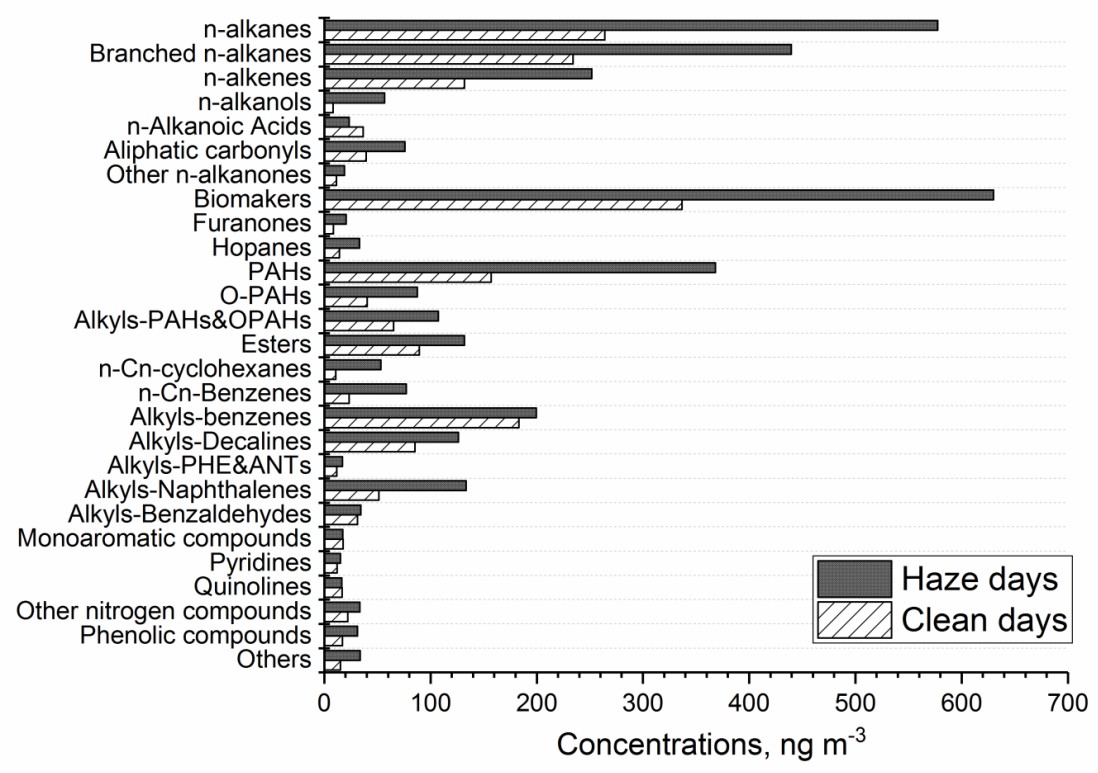


**Figure 2:** A comparison of organic compound groups between non-haze and haze days. The average total concentration of the identified group was calculated in the non-haze (13 days) and haze periods (20 days), respectively.






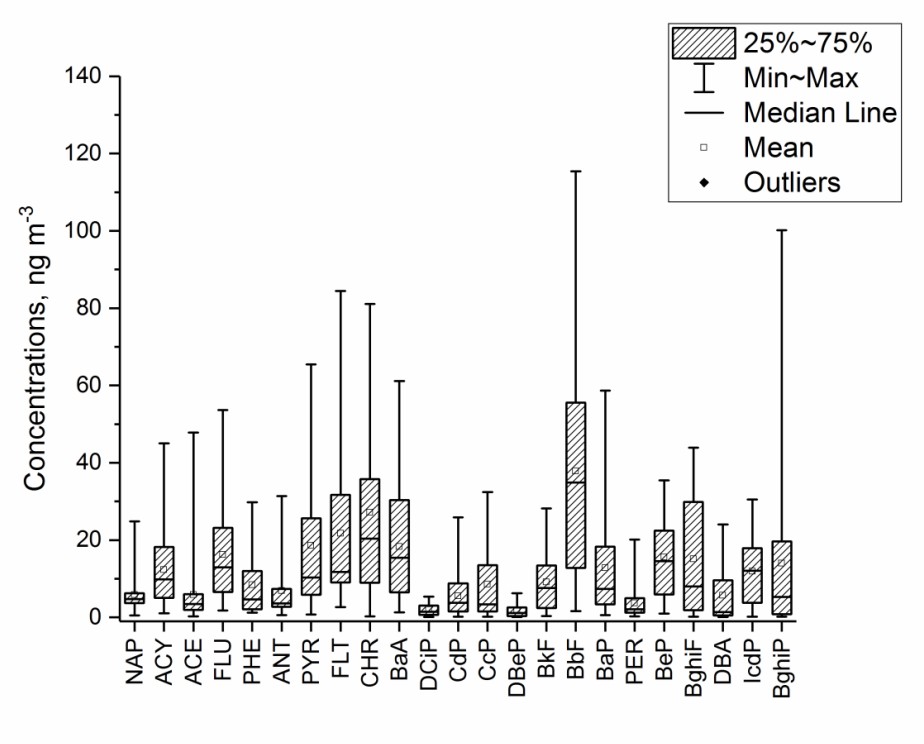


**Figure 3:** The distribution of concentrations of PAHs.






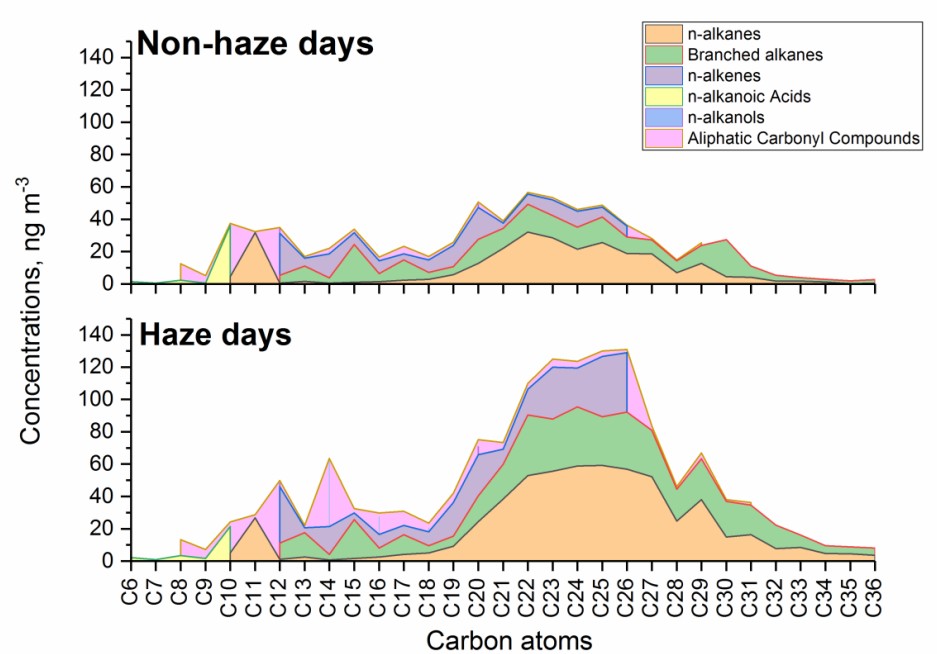


**Figure 4:** The molecular distributions of aliphatic hydrocarbons and other homologous series, including n-alkanes, branched alkanes, n-alkenes, carbonyl compounds (n-alkanals, n-alkan-2-ones, n-alkan-3-ones), n-alkanoic acid and alkanols on haze and non-haze days.







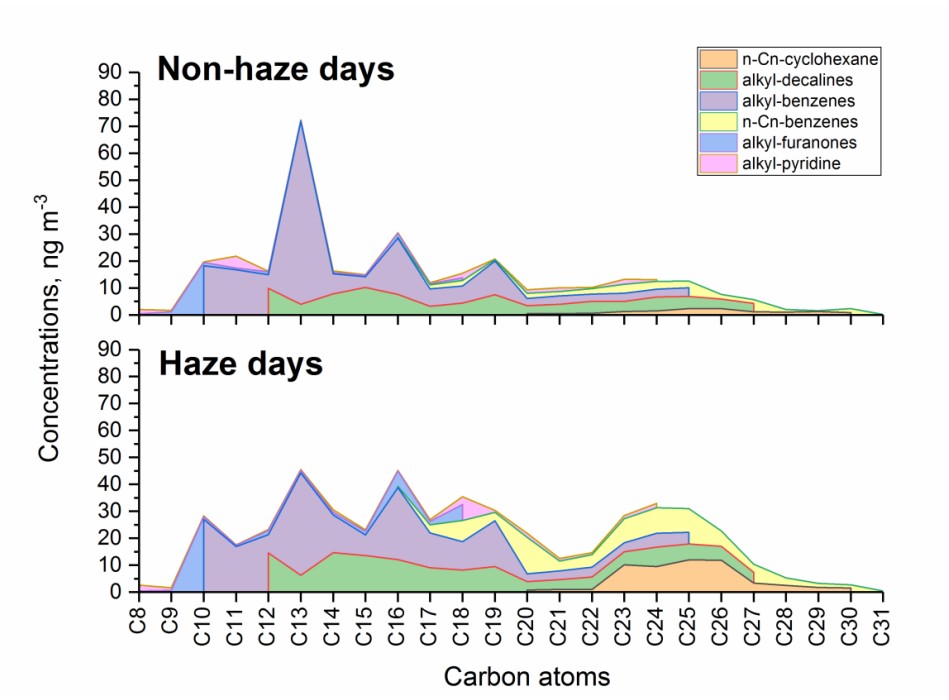


**Figure 5:** The molecular distributions of n-$C_n$-cyclohexane, alkyl-bicyclic-alkanes, alkyl-benzenes, n-$C_n$-benzenes, alkyl-furanones and alkyl-pyridines on haze and non-haze days.






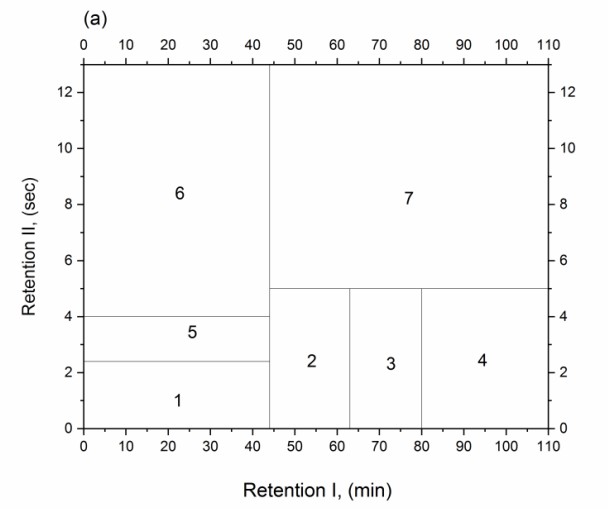


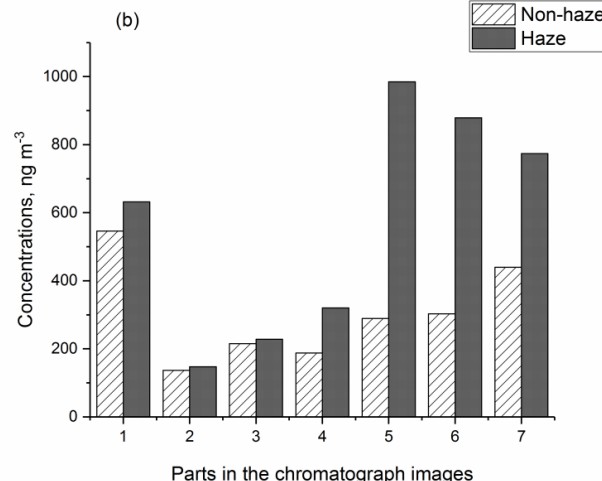


**Figure 6:** The separated chromatogram from the GC × GC-TOFMS. (a) the diagram of the separated
image with seven parts; (b) The concentration results of unknown organic compounds in each
chromatography image part during non-haze and haze days.
