# Peer review of "Insight into the Composition of Organic Compounds"

_Atmospheric Chemistry and Physics, 2018_

## Short Comment (SC1) · 12 Feb 2019

Interesting data are presented in this paper. Over 300 compounds were found in aerosols collected during haze and non haze events. Many chemical classes of compounds were identified, which there are few papers discussing them. Excellent information for atmospheric chemists. If more samples were collected, PMF could show the contribution of the emissions sources.

---

## Referee Comment (RC1) · Anonymous Referee #1 · 1 Mar 2019

General comments: This paper described the characteristics of organic compounds (>C6) in PM2.5 from Beijing during wintertime. More than 300 organic compounds, accounting for approximately 47% of the total organic compounds in the chromatogram, were detected by GC×GC-TOFMS. The overall strength of this study is acquisition of a detailed dataset of organic compounds taken over one-month period that spanned non-haze and haze days. The topic of the paper is well suited for ACP, and the data itself are interesting. On the whole, English language requires substantial improvement throughout the manuscript. Many sentences are not clearly written, leaving the reader puzzling about their meaning. In addition, the overall weakness is the data interpretation. More effort needs to be put into presentation of the results. I have some points

where more information is needed or where I disagree.

Specific comments: 1. Introduction: The characterization and source identification of organic compounds in PM in Beijing have been extensively studied. I would suggest authors to improve the introduction by summarizing these previous studies and providing some results in line with the major conclusion of this study.

2. Line 70: A reference here, regarding the number "96 ug m-3" was taken, would be helpful.

3. Line 73 and throughout the paper: Please give a definition of PAH species at their first appearance in the text, and then the abbreviation should be used. Nomenclature for PAH or PAHs should keep consistent throughout the paper.

4. Line 75: Please clarify the importance of group type used in this study.

5. Line 80: What does "three independent analytical dimensions" here? Not two-dimensions?

6. Line 95: The objective of this study is to identify the sources and formation processes of the organic compounds. Is there any new findings on them which can not be obtained by conventional method? Please clarify in the text somewhere.

7. Line 105: Please check the location of the sampling site. 89o58'28"N, 11o62'16"E is right?

8. Lines 167 and 168: "4 mean concentrations within 18%, 6 within 10-20%", here what are the difference between 18% and 10-20%?

9. Lines 175-185: I would suggest the authors to give a general characteristics of pollutants and meteorological conditions during haze and non-haze days, respectively, since the authors focus on the comparison between the characteristics of organic groups on non-haze and haze days.

10. Line 203: Here, the authors cited the study in Nanjing. It would be better if the

authors compare their results with the previous studies in Beijing.

11. Line 209: Haze has been defined in Line 176. Here, consider deleting the definition.

12. Line 221: Table 1 shows the comparison of identified organic compounds between the present and previous studies in Beijing. We can see the big differences. Is it possible that the differences may arise from the differences in analytical techniques? or meteorological conditions? Additionally, can the authors speculate something about the data in this table? The n-alkanes make the greatest contribution to the identified OM. Why no alkanes in this table? Why DBP, EDP and so on are classified into the groups of Alkylated-PAHs, and Ester again? The concentrations of phenolic compounds are up to 2739 ng m-3? It is impossible in my opinion.

13. Section 3.3: The authors compare the characteristics of five organic compound groups on haze and non-haze days. I feel this section cloud be greatly improved. I suggest they focus on the more conclusive finding of this study rather than the previous studies. I use the part of n-alkanoic acids to illustrate my main concerns of this section. They have gone to the previous studies too many words, from Lines 224-243. Only two sentences described the present results. And the authors state consistent results for the acids were observed in this study. If so, how is the different? how is the consistent? The concentration levels or the distribution or whatever? From my opinion, they are significantly different in concentrations. I personally suggest to consider these differences and provide an in-depth insights into them. Additionally, section 3.3.1 title includes alkanones, but I can not see any description about them in this section. On the whole, I would recommend rewording Section 3.3 to focus on the new findings of this study.

14. Lines 429-431: The authors attribute the similar increase of n-alkanes and branched alkanes to a common source. I am not really able to follow what authors mean. Does that mean they are from the same source? This needs to be properly

explained. Not only the sources but also the atmospheric processes that could lead to the similar changes.

15. Section 3.5: I think this section is the novelty of this study. I believe that the tables represented in supplement appear to be more important. So I would suggest bringing some supplementary tables to the main text and proving more discussion in this section.

16. Line 464: A reference would be helpful.

17. Section 3.6: I feel authors draw mostly speculative conclusions in this section. It is not surprised to observe he complex physical and chemical processes of ambient aerosols. In fact, many researches have reported these complexities already. Then, what is the new finding of this study? Please clarify.

18. Figures 1 & 2: I am not able to follow what authors mean. What is the difference OPAHs in "O-PAHs" and "Alkyls-PAHs &OPAHs"?

---

## Referee Comment (RC2) · Anonymous Referee #2 · 29 Mar 2019

General comments:

This manuscript presents the quantification of organic compounds (> C6) in PM2.5 aerosol samples collected in Beijing in wintertime of 2016 using GC×GC-TOFMS technique. More than 300 organic compounds were determined and they were classified into different groups for discussion. The dataset of the identified organic compounds in this study is very interesting and useful for atmospheric research. However, the data interpretation is not well organized and suggested to be improved. In addition, as many sentences in the text is not easy to understand, the English language needs to be modified. I think this manuscript would make a nice contribution to the literature if the following comments can be addressed.

Major comments:

1. Section 3.3 'The charateistics…' is highly suggested to be improved in terms of data interpretation and English language.
   The majority of words in the section was written for the data presentation of previous referenced studies, while only a few sentences were used to interpret the data observed in current study. I think the authors should describe/interpret the current data more in detail.

2. I'm wondering the conclusion in line 49 in the abstract stating that 'organic aerosol is more highly oxidized and henxe less GC-volatile on haze days'. Please see the comments 11 and 15 in the specific comments.

Specific comments:

1. Line 34 and throughout the manuscript: in some sections, 'organic compounds (> C6)' is used, while in other sections 'organic compounds (≥C6)' is used. Please keep consistent throughout the manuscript.

2. Line 40 and throughout the manuscript: the full name of PAHs should be given at their first appearance in the manuscript and the abbreviation should be used in the following text. Meanwhile, the abbreviation of PAHs or PAH should keep consistent throughout the manuscript.

3. Line 43-44: the sentence 'A near-unimodal…in most hydrocarbon groups' is not understandable. Maybe you could rephrase 'the most abundant of hydrocarbon groups were observed with carbon atom range of 19-28' or something like that.

4. Line 106: The Figure 1 showing the sampling site is not found in the manuscript.

5. Section 2.2: Please generally state the analytical method in this manuscript, even it has been described in previous publication.

6. Line 125: 'polycyclic aromatic hydrocarbons (PAHs)', the abbreviation of PAHs should be used instead of the full name.

7. Line 124-136: please check the number of standards used in this study. For example, '6 alkyl-benzenes (…)'. However, there are only 5 standard compounds in the bracket. '15 alkyl-cyclohexanes (…)' should only include 13 standard compounds?

8. Line 155: please define the full name of 'IS'

9. Line 167-168: what do the values of 18%, 20-30%..mean? Do they show the difference of concentrations observed in GC*GC method and conventional GC method?

10. Line 223: The tile of section 3.3.1 shows the short chain fatty acids….and alkanones. However, I did not see any result or discussion of short chain fatty acids and alkanones.

11. Line 243-246: It shows that higher concentration of alkanoic acids was observed on non-haze days compared to that on haze days. Does it indicate that organic compounds in non-haze days experience more intensive oxidation process? However, this indication disagrees with the main conclusion in the Abstract, showing that organic aerosols is more highly oxidized on haze days.

12. Line 328-329: why the concentration of O-PAHs was calculated in both Σ O-PAHs and Σ alkylated-PAHs and O-PAHs?

13. Line 932: Figure 3, what does the dashed bar of 25-75% mean?

14. Line 335: I did not see the concentration of retene in Figure 3.

15. Line 356-357: Lower ratios of quinone: parent PAH were observed on haze days compared to non-haze days. According to this observation, I might think that organic compounds on non-haze days were highly oxidaized, which is opposite with the author's conclusion in the Abstract (also see the comment 11).

16. Line 417-418: the sentence 'there is a clear…' is not understandable, please rephrase it.

17. Line 420-421: In Figure 4, the contribution of C19-C28 compounds to the total identified organic compounds looks similar between haze days and non-haze days. Why do you state that a higher contribution was observed in haze days? Could you please give the values of their contributions?

18. Line 494-495: the sentence 'a low ratio…' is difficult to understand. What does the 'low ratio' mean?

---

## Author Comment (AC1) · 26 Jun 2019

Journal: ACP MS No.: acp-2018-1273 Title: Insight into the Composition of Organic Compounds ($\geq$ C6) in PM2.5 in Wintertime in Beijing, China Author(s): Ruihe Lyu et al. Special Issue: In-depth study of air pollution sources and processes within Beijing and its surrounding region (APHH-Beijing) (ACP/AMT inter-journal SI)

RESPONSE TO REVIEWERS

REFEREE #1 General comments: This paper described the characteristics of organic compounds (>C6) in PM2.5 from Beijing during wintertime. More than 300 organic

compounds, accounting for approximately 47% of the total organic compounds in the chromatogram, were detected by GC×GC-TOFMS. The overall strength of this study is acquisition of a detailed dataset of organic compounds taken over one-month period that spanned non-haze and haze days. The topic of the paper is well suited for ACP, and the data itself are interesting. On the whole, English language requires substantial improvement throughout the manuscript. Many sentences are not clearly written, leaving the reader puzzling about their meaning. In addition, the overall weakness is the data interpretation. More effort needs to be put into presentation of the results. I have some points where more information is needed or where I disagree.

Specific comments: 1. Introduction: The characterization and source identification of organic compounds in PM in Beijing have been extensively studied. I would suggest authors to improve the introduction by summarizing these previous studies and providing some results in line with the major conclusion of this study. RESPONSE: The Introduction has been restructured.

2. Line 70: A reference here, regarding the number "96 ug m-3" was taken, would be helpful. RESPONSE: The data has been updated, and a reference added (Li et al., 2019).

Li, L. J., Ho, S. S. H., Feng, B., Xu, H., Wang, T., Wu, R., Huang, W., Qu, L., Wang, Q., and Cao, J.: Characterization of particulate-bound polycyclic aromatic compounds (PACs) and their oxidations in heavy polluted atmosphere: A case study in urban Beijing, China during haze events, Sci. Tot. Environ., 660, 1392-1402, 2019.

3. Line 73 and throughout the paper: Please give a definition of PAH species at their first appearance in the text, and then the abbreviation should be used. Nomenclature for PAH or PAHs should keep consistent throughout the paper. RESPONSE: This has been corrected.

4. Line 75: Please clarify the importance of group type used in this study. RESPONSE: The importance of group has been clarified in the manuscript and a reference has been

added (Alam et al. 2016).

Alam, M. S., Stark, C., and Harrison, R. M.: Using variable ionisation energy time-of-flight mass spectrometry with comprehensive GC×GC to identify isomeric species, Anal. Chem., 88, 4211-4220, 2016.

5. Line 80: What does "three independent analytical dimensions" here? Not two dimensions? RESPONSE: Corrected as two dimensions.

The following figure (GCxGC-TOFMS image) shows the three dimensions:

First dimension: volatility; Second dimension: polarity; Third dimension: response (signal) of organic compound;

6. Line 95: The objective of this study is to identify the sources and formation processes of the organic compounds. Is there any new findings on them which cannot be obtained by conventional method? Please clarify in the text somewhere. RESPONSE: This has been clarified. The greater resolution in the chromatography allows measurement of a greater number of compounds in the same air samples, allowing greater in-depth analysis. We have also analysed the distribution of compounds across the chromatogram in terms of their volatility and polarity (Section 3.5) revealing major differences between haze and non-haze days which would not be possible with conventional chromatographic methods.

7. Line 105: Please check the location of the sampling site. 89o58'28"N, 11o62'16"E is right? RESPONSE: The location information has been corrected.

8. Lines 167 and 168: "4 mean concentrations within 18%, 6 within 10-20%", here what are the difference between 18% and 10-20%? RESPONSE: The description has been revised.

9. Lines 175-185: I would suggest the authors to give a general characteristics of pollutants and meteorological conditions during haze and non-haze days, respectively, since the authors focus on the comparison between the characteristics of organic groups on

non-haze and haze days. RESPONSE: We have added descriptions of characteristics of pollutants and meteorological conditions as suggested. The following reference has been added (Lyu et al., 2019).

Lyu, R., Shi, Z., Alam, M. S., Wu, X., Liu, D., Vu, T. V., Stark, C., Fu, P., Feng, Y., and Harrison, R. M.: Alkanes and aliphatic carbonyl compounds in wintertime PM2.5 in Beijing, China, Atmos. Environ., 202, 244-255, 2019.

10. Line 203: Here, the authors cited the study in Nanjing. It would be better if the authors compare their results with the previous studies in Beijing. RESPONSE: We believe that the Nanjing data are useful, and have compared them with Beijing. The following reference has been added (Haque et al., 2019).

Haque, M., Kawamura, K., Deshmukh, D. K., Fang, C., Song, W., Mengying, B., and Zhang, Y.-L.: Characterization of organic aerosols from a Chinese megacity during winter: predominance of fossil fuel combustion, Atmos. Chem. Phys., 19, 5147-5164, 2019.

11. Line 209: Haze has been defined in Line 176. Here, consider deleting the definition. RESPONSE: Deleted.

12. Line 221: Table 1 shows the comparison of identified organic compounds between the present and previous studies in Beijing. We can see the big differences. Is it possible that the differences may arise from the differences in analytical techniques? or meteorological conditions? Additionally, can the authors speculate something about the data in this table? The n-alkanes make the greatest contribution to the identified OM. Why no alkanes in this table? Why DBP, EDP and so on are classified into the groups of Alkylated-PAHs, and Ester again? The concentrations of phenolic compounds are up to 2739 ng m-3? It is impossible in my opinion. RESPONSE: We thank the reviewer for raising these important questions. The differences in concentration may arise from the analytical methods, with GC-MS liable to overestimate the concentrations of organic compounds due to the very high baseline caused by the UCM. The extract solutions in

the present study were not subject to derivatization, and this may have caused the loss of some alcohols and acids.

The detailed data for n-alkanes is in the Lyu et al. (2019) paper cited within the manuscript. More discussion of n-alkanes has been added.

DBP and DEP have been deleted from the group of PAHS.

The units are wrong, and should be pg m-3; this has been corrected in the table.

13. Section 3.3: The authors compare the characteristics of five organic compound groups on haze and non-haze days. I feel this section cloud be greatly improved. I suggest they focus on the more conclusive finding of this study rather than the previous studies. I use the part of n-alkanoic acids to illustrate my main concerns of this section. They have gone to the previous studies too many words, from Lines 224-243. Only two sentences described the present results. And the authors state consistent results for the acids were observed in this study. If so, how is the different? how is the consistent? The concentration levels or the distribution or whatever? From my opinion, they are significantly different in concentrations. I personally suggest to consider these differences and provide an in-depth insights into them. Additionally, section 3.3.1 title includes alkanones, but I cannot see any description about them in this section. On the whole, I would recommend rewording Section 3.3 to focus on the new findings of this study. RESPONSE: This section has been significantly restructured.

The detailed data for n-alkanones is in the Lyu et al. (2019) paper which we cite, and the carbonyl compounds (n-alkanals, alkanones) are now described in Section 3.3.1.

14. Lines 429-431: The authors attribute the similar increase of n-alkanes and branched alkanes to a common source. I am not really able to follow what authors mean. Does that mean they are from the same source? This needs to be properly explained. Not only the sources but also the atmospheric processes that could lead to the similar changes. RESPONSE: The wording has been changed to clarify this point.

The similar behaviour implies that n-alkanes and branched alkanes arise either from the same source, or from sources with highly correlated emissions.

15. Section 3.5: I think this section is the novelty of this study. I believe that the tables represented in supplement appear to be more important. So I would suggest bringing some supplementary tables to the main text and proving more discussion in this section. RESPONSE: This section has been restructured to reflect the suggestion of the reviewer and a new table added to the text.

16. Line 464: A reference would be helpful. RESPONSE: A reference (Cao et al., 2018) has been added.

Cao, R., Zhang, H., Geng, N., Fu, Q., Teng, M., Zou, L., Gao, Y., and Chen, J.: Diurnal variations of atmospheric polycyclic aromatic hydrocarbons (PAHs) during three sequent winter haze episodes in Beijing, China, Sci. Tot. Environ., 625, 1486-1493, 2018.

17. Section 3.6: I feel authors draw mostly speculative conclusions in this section. It is not surprised to observe he complex physical and chemical processes of ambient aerosols. In fact, many researches have reported these complexities already. Then, what is the new finding of this study? Please clarify. RESPONSE: This section provides a discussion of possible general conclusions deriving from the data. The messages drawn from the data are not wholly consistent with one another, and additional text seeks to clarify this point as far as possible.

18. Figures 1 & 2: I am not able to follow what authors mean. What is the difference OPAHs in "O-PAHs" and "Alkyls-PAHs & OPAHs"? RESPONSE: These are alkylated-PAHs and alkylated-OPAHs.

REVIEWER #2 General comments: This manuscript presents the quantification of organic compounds (> C6) in PM2.5 aerosol samples collected in Beijing in wintertime of 2016 using GC×GC-TOFMS technique. More than 300 organic compounds were

determined and they were classified into different groups for discussion. The dataset of the identified organic compounds in this study is very interesting and useful for atmospheric research. However, the data interpretation is not well organized and suggested to be improved. In addition, as many sentences in the text is not easy to understand, the English language needs to be modified. I think this manuscript would make a nice contribution to the literature if the following comments can be addressed.

Major comments: 1. Section 3.3 'The charateistics...' is highly suggested to be improved in terms of data interpretation and English language. The majority of words in the section was written for the data presentation of previous referenced studies, while only a few sentences were used to interpret the data observed in current study. I think the authors should describe/interpret the current data more in detail. RESPONSE: This section has been revised as recommended. However, there remains much previous literature with which to compare and hence the length.

2. I'm wondering the conclusion in line 49 in the abstract stating that 'organic aerosol is more highly oxidized and henxe less GC-volatile on haze days'. Please see the comments 11 and 15 in the specific comments. RESPONSE: There are apparent anomalies in the data which we have done our best to explain (see responses to comments 11 and 15).

Specific comments: 1. Line 34 and throughout the manuscript: in some sections, 'organic compounds (> C6)' is used, while in other sections 'organic compounds (≥C6)' is used. Please keep consistent throughout the manuscript. Response: This has been corrected in the manuscript to be consistent.

2. Line 40 and throughout the manuscript: the full name of PAHs should be given at their first appearance in the manuscript and the abbreviation should be used in the following text. Meanwhile, the abbreviation of PAHs or PAH should keep consistent throughout the manuscript. Response: This has been corrected in the manuscript.

3. Line 43-44: the sentence 'A near-unimodal...in most hydrocarbon groups' is not

understandable. Maybe you could rephrase 'the most abundant of hydrocarbon groups were observed with carbon atom range of 19-28' or something like that. Response: This has been corrected in the manuscript. 4. Line 106: The Figure 1 showing the sampling site is not found in the manuscript. Response: This is now included.

5. Section 2.2: Please generally state the analytical method in this manuscript, even it has been described in previous publication. Response: The analytical method is described in the Section 2.3

6. Line 125: 'polycyclic aromatic hydrocarbons (PAHs)', the abbreviation of PAHs should be used instead of the full name. Response: Corrected

7. Line 124-136: please check the number of standards used in this study. For example, '6 alkyl-benzenes (...)'. However, there are only 5 standard compounds in the bracket. '15 alkyl-cyclohexanes (...)' should only include 13 standard compounds? Response: This has been corrected

8. Line 155: please define the full name of 'IS' Response: Defined, it is Internal Standard.

9. Line 167-168: what do the values of 18%, 20-30%..mean? Do they show the difference of concentrations observed in GC*GC method and conventional GC method? Response: Yes, this is now clarified in the text.

10. Line 223: The tile of section 3.3.1 shows the short chain fatty acids....and alkanones. However, I did not see any result or discussion of short chain fatty acids and alkanones. Response: Corrected, and the carbonyl compounds (n-alkanals, alkanones) are now described in Section 3.3.1.

11. Line 243-246: It shows that higher concentration of alkanoic acids was observed on non-haze days compared to that on haze days. Does it indicate that organic compounds in non-haze days experience more intensive oxidation process? However, this indication disagrees with the main conclusion in the Abstract, showing that organic

aerosols is more highly oxidized on haze days. Response: The alkanoic acids are thought to mainly originate from cooking, and the data are consistent with the report of Sun (Sun et al., 2013), that cooking makes a larger contribution to the OA in the non-haze days.

12. Line 328-329: why the concentration of O-PAHs was calculated in both $\Sigma$ O-PAHs and $\Sigma$ alkylated-PAHs and O-PAHs? Response: The "alkylated-PAHs and O-PAHs" means alkylated-PAHs and alkylated O-PAHs, and has been replaced with Alkylated-(PAHs & OPAHs).

13. Line 932: Figure 3, what does the dashed bar of 25-75% mean? Response: 25%-first quartile, 75%-third quartile. This explanation has been added in the title of Figure 3.

14. Line 335: I did not see the concentration of retene in Figure 3. Response: The retene was classified into alkylated-(PAHs & OPAHs)

15. Line 356-357: Lower ratios of quinone: parent PAH were observed on haze days compared to non-haze days. According to this observation, I might think that organic compounds on non-haze days were highly oxidaized, which is opposite with the author's conclusion in the Abstract (also see the comment 11). Response: We also found this surprising and conclude that the low ratios probably demonstrate that the oxidation processes continue leading to formation of other compounds. We note that Li et al. (2019) found no difference in ïÅŞOPAH to ïÅŞPAH ratios between haze and clean air periods in Beijing, consistent with our data.

16. Line 417-418: the sentence 'there is a clear. . .' is not understandable, please rephrase it. Response: Corrected. 17. Line 420-421: In Figure 4, the contribution of C19-C28 compounds to the total identified organic compounds looks similar between haze days and non-haze days. Why do you state that a higher contribution was observed in haze days? Could you please give the values of their contributions? Response: The contributions are now stated in the manuscript.

18. Line 494-495: the sentence 'a low ratio...' is difficult to understand. What does the 'low ratio' mean? Response: New wording clarifies which ratio is referred to. The ratios between haze and non-haze days have been added into Table S3.
* * *

---

## Referee Report (RR1)

Comments:

This manuscript has been well improved and the authors have answered most of the questions. However, I still have one question regarding the conclusion of 'There is strong evidence that the organic aerosol is more highly oxidized ,...on haze days' in the abstract (Line 49-50).

In line 534-537 'By definition, concentrations of $PM_{2.5}$ are elevated during haze events, but the question arises as to whether primary or secondary organic compounds make a larger contribution to the rise in concentrations. Constituents that are expected to be primary are typically elevated in mean concentration by a factor of around two', it shows the higher primary contribution on haze days compared to non-haze days. In Line 376 'lower ratios of O-PAHs/PAHs on haze days than non-haze days were observed'. In line 549-554 'This result was consistent with Section 3.5; ...a low ratio may reflect a high degree of further processing to form more oxidised 553 species on the haze days compensating for enhanced formation'. These observation or discussion probably indicates that the organic aerosol on non-haze days are more oxidized. So, I'm doubting about the conclusion in the abstract.

If this comment can be addressed, I am very happy to suggest this manuscript to be published in Atmos. Chem. Phys.

---

## Author Response (AR3)

**Journal: ACP**
**MS No.: acp-2018-1273**
**Title: Insight into the Composition of Organic Compounds (≥ C6) in PM2.5 in Wintertime in Beijing, China**
**Author(s): Ruihe Lyu et al.**
**Special Issue: In-depth study of air pollution sources and processes within Beijing and its surrounding region (APHH-Beijing) (ACP/AMT inter-journal SI)**

**RESPONSE TO REVIEWER #2**

Comments: This manuscript has been well improved and the authors have answered most of the questions. However, I still have one question regarding the conclusion of 'There is strong evidence that the organic aerosol is more highly oxidized ,…on haze days' in the abstract (Line 49-50). In line 534-537 'By definition, concentrations of PM2.5 are elevated during haze events, but the question arises as to whether primary or secondary organic compounds make a larger contribution to the rise in concentrations. Constituents that are expected to be primary are typically elevated in mean concentration by a factor of around two', it shows the higher primary contribution on haze days compared to non-haze days. In Line 376 'lower ratios of O-PAHs/PAHs on haze days than non-haze days were observed'. In line 549-554 'This result was consistent with Section 3.5; …a low ratio may reflect a high degree of further processing to form more oxidised 553 species on the haze days compensating for enhanced formation'. These observation or discussion probably indicates that the organic aerosol on non-haze days are more oxidized. So, I'm doubting about the conclusion in the abstract. If this comment can be addressed, I am very happy to suggest this manuscript to be published in Atmos. Chem. Phys.

**RESPONSE:** We thank the reviewer for pointing out the apparent inconsistency. The statement in the abstract is based upon the distribution of compounds in the chromatogram, and the final sentence of the abstract has been amended to reflect this more clearly.

[revised manuscript text omitted]

➢ carbon numbers (n-alkanes) ≤ 17;
➢ monoaromatics; | 802 | 546 | 911 | 632 |
| 2 | Medium molecular weight:
➢ 17 < carbon numbers (n-alkanes) ≤ 23;
➢ Oxidized hydrocarbons (alkanals, alkanones); | 334 | 137 | 483 | 147 |
| 3 | Medium molecular weight:
➢ 23 < carbon numbers (n-alkanes) ≤ 27;
➢ Oxidized hydrocarbons (alkanals, alkanones); | 573 | 215 | 1060 | 228 |
| 4 | High molecular weight:
➢ carbon numbers (n-alkanes) ≥ 27; | 351 | 188 | 730 | 320 |
| 5 | Oxidized monoaromatics; | 621 | 289 | 1309 | 985 |
| 6 | 2 rings PAHs | 485 | 303 | 1556 | 879 |
| 7 | 3-6 rings PAHs and hopanes; | 792 | 440 | 1337 | 774 |
| | **Total** | 3958 | 2119 | 7385 | 3964 |

[Figure]

**Figure 1:** The percentages of the organic compound groups in the total identified organic compounds.

[Figure]

**Figure 2:** A comparison of organic compound groups between non-haze and haze days. The average
total concentration of the identified group was calculated in the non-haze (13 days) and haze periods
(20 days), respectively.

[Figure]

**Figure 3:** The distribution of concentrations of PAHs (shaded bars, 25%-first quartile, 75%-third
quartile).

[Figure]

**Figure 4:** The molecular distributions of aliphatic hydrocarbons and other homologous series, including n-alkanes, branched alkanes, n-alkenes, carbonyl compounds (n-alkanals, n-alkan-2-ones, n-alkan-3-ones), n-alkanoic acid and alkanols on haze and non-haze days.

[Figure]

**Figure 5:** The molecular distributions of n-$C_n$-cyclohexane, alkyl-bicyclic-alkanes, alkyl-benzenes, n-
$C_n$-benzenes, alkyl-furanones and alkyl-pyridines on haze and non-haze days.

[Figure]

**Figure 6:** The concentration (ng m$^{-3}$) sum of identified and unknown organic compounds in each chromatogram image section during (a) non-haze and (b) haze days.